# Deciphering Rhizosphere Microbiome Assembly of *Castanea henryi* in Plantation and Natural Forest

**DOI:** 10.3390/microorganisms10010042

**Published:** 2021-12-26

**Authors:** Yuanyuan Cheng, Lexin Zhou, Tian Liang, Jiayin Man, Yinghao Wang, Yu Li, Hui Chen, Taoxiang Zhang

**Affiliations:** 1Oil Tea Research Center of Fujian Province, College of Forestry, Fujian Agriculture and Forestry University, Fuzhou 350002, China; yuanyycheng@163.com (Y.C.); ZLX2762190752@163.com (L.Z.); liangt_forest@163.com (T.L.); yulitrees@163.com (Y.L.); 2College of Forestry, Fujian Agriculture and Forestry University, Fuzhou 350002, China; jiayinmanjiayin@163.com (J.M.); 18438616077@163.com (Y.W.); 3International Joint Laboratory of Forest Symbiology, College of Forestry, Fujian Agriculture and Forestry University, Fuzhou 350002, China

**Keywords:** natural and plantation *Castanea henryi* forests, rhizosphere microorganism, molecular ecological network

## Abstract

Based on the importance and sensitivity of microbial communities to changes in the forest ecosystem, soil microorganisms can be used to indicate the health of the forest system. The metagenome sequencing was used to analyze the changes of microbial communities between natural and plantation *Castanea henryi* forests for understanding the effect of forest types on soil microbial communities. Our result showed the soil microbial diversity and richness were higher in the natural forests than in the plantation. *Proteobacteria*, *Actinobacteria*, and *Acidobacteria* are the dominant categories in the *C*. *henryi* rhizosphere, and *Proteobacteria* and *Actinobacteria* were significantly enriched in the natural forest while *Acidobacteria* was significantly enriched in the plantation. Meanwhile, the functional gene diversity and the abundance of functions in the natural forest were higher than that of the plantation. Furthermore, we found that the microbial network in the natural forests had more complex than in the plantation. We also emphasized the low-abundance taxa may play an important role in the network structure. These results clearly showed that microbial communities, in response to different forest types, provide valuable information to manipulate microbiomes to improve soil conditions of plantation.

## 1. Introduction

Soil microorganisms are an important part of the soil ecosystem [1,2,3]. The diversity of soil microbial composition and function play a critical role in maintaining soil productivity and stability, such as nutrient cycling [4,5], and pollutant degradation [6,7]. Meanwhile, rhizosphere microbiota can largely determine plant metabolism and physiological activities, including driving nutrient acquisition [8,9], regulating plant growth [10], and protecting hosts from biological and abiotic stresses [11,12,13].

Soil microbial community structure was reported to be sensitive to changes in soil environment. Due to the complexity of forest soil ecosystem, vegetation types [14,15], fertilization [16,17], irrigation [18,19], and land-use type [20] have a great impact on the number and species of soil microorganisms. For instance, the conversion of natural forest to poplar forest plantation reduced the organic matter content and humidity values and lowered the diversity of soil microbial communities [21]. After the natural forest was transformed into rubber plantation, the content of soil organic matter and total nitrogen decreased and the abundance of *Actinomycetes*, arbuscular mycorrhizal fungi, fungi, bacteria and protozoa decreased significantly [22]. Meanwhile, the tree species could influence soil microbial community [23,24]. Studies had shown that fungal diversity increased with the increase in tree diversity [25,26]. In addition, vegetation types can also affect the richness and activity of soil microorganisms by plant litter. Cong et al. [14] considered that significant differences in soil microbial communities among three vegetation types (a coniferous forest, a mixed broadleaf forest, and a deciduous broadleaf forest) were probably explained by decomposing the plant litter with different chemical structures. In view of the sensitivity of soil microbial community structure and function to environmental changes, the characteristics of soil microbial community structure can be used as one of the important indicators of soil quality changes [27,28].

*Castanea henryi*, an important economic species, has a long history of artificial cultivation and is mainly distributed in Zhejiang, Jiangxi, Hunan, and Fujian, among which Fujian has the largest planting area and yield in the country [29,30]. With the continuous expansion of artificial *C. henryi* planting regions, the area of the remaining natural forest has dropped sharply, which was half of that of plantation forest [31]. The natural forest has complex community structure and plant species, while the plantation is characterized by intensive monoculture [32]. The amount of litter returned in plantation is significantly lower than that in natural forest [33,34,35]. In addition, long-term fertilization and spraying pesticides in plantation have led to soil acidification, consolidation, soil erosion, and organic pollution [36,37], which would destroy the dynamic balance of soil microbial community [38,39]. According to the statistics, in 2016, the area of soil erosion of *C. henryi* forest in Songxi County, Fujian Province, was 665.8 hectares, accounting for 90% of the area of *C. henryi* forest [40]. A large number of researchers showed that soil erosion and soil degradation can seriously break the dynamic balance of soil microbial community, and the imbalance of soil microbial community will further aggravate soil degradation [41,42,43]. Therefore, the study on the structure and functional diversity of soil microbial community is of great significance to understand the soil quality of artificial *C. henryi* forest. Meanwhile, it could provide guidance in preventing soil fertility decline and scientific managing of plantation *C. henryi* forest.

At present, the research of *C. henryi* plantation mainly focused on the development and utilization of germplasm resources, genetic diversity, high-yield cultivation techniques, and occurrence and control of main diseases and pests [29,44,45]. However, the research on soil microbial community of *C. henryi* plantation is less investigated. In this study, metagenomic sequencing technology with microbial molecular ecological networks were combined to (1) characterize the change of composition and function of rhizosphere microbial community assembly at different forest types and cultivated varieties, (2) analyze the soil fertility degradation from the perspective of microorganisms, and (3) elucidate the effect of forest types on microbial molecular ecological networks. Here, we hypothesize that the monoculture and unreasonable management of the plantation may reduce the species and functional diversity of the microbial community and exacerbate competition among microorganisms to nutrients. The results of this research were expected to provide a scientific basis for the sustainable management of the plantation.

## 2. Materials and Methods

### 2.1. Sample Collection

The study area is located in Taining County (26° N, 117° E), Fujian Province, China (Appendix A). It belongs to a typical Central Asian hot monsoon climate, with a mean annual temperature of 13–23 ℃ and precipitation of 1,176 mm. There are a large number of natural and plantation *C. henryi* forests. The plantation was a mature 19 years old (in 2000) Castanea henryi forest and continuously subject to agricultural practice. Fertilizer was applied once per year by local farmers, with 1363 kg ha^−1^ farmyard manure, 545 kg ha^−1^ nitrogen fertilizer, and 273 kg ha^−1^.Meanwhile, propiconazole and glyphosate were used to manage plant diseases and weeding. In 2019, three different cultivated varieties “Chushuhong” (TRC), “Youzheng (TRY)”, and “Bailuzi”(TRB) of *C. henryi* plantation and natural *C. henryi* forest (TTX) with the same site conditions were selected as the research objects. The area of four sampling points is all greater than 1 hectare and the distance between the sampling points is not less than 200 m. Five standard sample plots (20 × 20 m) are randomly set at each of the four sampling points, and the distance between each sample plot is greater than 10 m. Weeds and topsoil were removed with a shovel to expose the root system. Rhizosphere soil was collected according to Zhang et al. [46]. Four soil samples (0–30 cm) were evenly collected and mixed into one rhizosphere soil, with a total of 20 rhizosphere soils. All the samples were transported back to the laboratory under low-temperature conditions. Stones and plant roots were removed and screened for 2 mm and then divided into two parts; one was stored at −80 °C for DNA extraction and the other was air-dried for the determination of soil physical and chemical properties.

Soil physical and chemical properties were analyzed according to Forestry Industry Standard of the People’s Republic of China. Briefly, soil moisture (SM) was calculated by drying method. Soil pH was determined using pH monitor (ST5000/B, OHAUS, Beijing, China) in a 1:2.5 soil to water (*w*/*v*) suspension. Organic matter (OM) content was determined by the principle of potassium dichromate oxidation-outer heating. A total 100 mg finely ground soil sample was wet digested to determine total phosphorus (TP) with nitric acid (HNO_3_), hydrofluoric acid (HF), and perchloric acid (HClO_4_) [47], while 1.0 g finely ground soil sample was digested to determine total potassium (TK) with HF–HClO_4_–HNO_3_ [48]. TP and TK contents were determined by inductively coupled plasma atomic emission spectrophotometry (ICP-AES) (PE OPTIMA 8000; America) after digestion. Soil total carbon (TC) and total nitrogen (TN) contents were determined by an elemental analyzer (vario MACRO; Elementar, Germany).

### 2.2. DNA Extraction and Metagenomic Data Analysis

DNA was extracted using CTAB method [49]. DNA concentrations and integrity determined by Qbiut3.0 and electrophoresis (1% agarose gel). Qualified DNA were used for the preparation of metagenomic library and sequencing by Biomarker Technology (Beijing, China). The experimental procedures were performed according to the standard protocol provided by Illumina. After passing the genomic DNA test, the DNA was fragmented by mechanical interrupt method (ultrasonic), then DNA fragments experienced purification, ends repair, adding A to 3′ ends, adding adaptors and PCR amplification to construct the sequencing library. After the library inspection was qualified, the qualified libraries were sequenced on the Illumina NovaSeq 6000 platform to obtain row data. Raw reads were filtered by Trimmomatic software (v0.33) and then aligned with host genome sequence by bowtie2 to identify and remove the host originated reads, so as to obtain high-quality sequencing data. Since the whole Genome of Castanea henry has not been published, we map these reads into the latest Chinese chestnut genes (https://github.com/yongshuai-sun/hhs-omei, accessed on 20 April 2021), which are a closely related species of Castanea henryi. Then, the contig sequences were filtered and assembled by megahit software to be evaluated by QUAST software. MetaGeneMark software (http://exon.gatech.edu/meta_gmhmmp.cgi, Version 3.26, accessed on 20 April 2021) was used to recognize coding regions of the genome with default parameters. The non-redundant metagenes were obtained by MMseqs2 software (https://github.com/soedinglab/mmseqs2, Version 12-113e3, accessed on 20 April 2021), with the similarity threshold of 95% and the coverage threshold of 90%. The species composition and relative abundance information of the samples were obtained according to the sequence of the non-redundant genes aligned to Nr. In order to obtain the annotation information of the corresponding gene, the non-redundant genes were BLAST with the sequence of recorded protein in KEGG database.

### 2.3. Construction of Molecular Ecological Network

Network construction and network property parameters are obtained in Molecular Ecological Network Analyses Pipeline (MENA) (http://ieg4.rccc.ou.edu/mena, accessed on 6 July 2021). The adjacency matrix was derived by applying the appropriate threshold value. We ranked all nodes in the network from high to low according to the degree and selected the top 10 nodes as the keystone taxa in the network [50]. Meanwhile, in order to clarify the relationship between species relative abundance and keystone taxa, we defined the level of species abundance following previous studies [51,52,53]. Briefly, abundant taxa were defined as the average relative abundance >0.1% within five repeated samples, whereas the average relative abundance of <0.01% were defined as rare taxa, and other abundance was defined as moderate taxa.

### 2.4. Statistical Analysis

All statistics tests and the different test data analyses were performed with SPSS (v26.0) and STAMP (v2.1.3) software. Kruskal–Wallis rank-sum test was used for multi-group comparison, and Welch’s *t*-test or Wilcoxon rank-sum test were used for pairwise comparison. Microbial diversity and richness were characterized by ACE, Chao1, Shannon, and Simpson indices. The relative abundances of main microbial species and function were visualized by R software (v4.0.3) and Python software (v3.9). The relationship between the microbial community and environmental factors was estimated by Spearman correlation. The molecular ecological network was visualized using the software of Cytoscape (v3.8.2).

## 3. Results

### 3.1. Diversity of Microbial Communities at Different Forest Types and Cultivated Varieties

The rhizosphere soils of 5 *C. henryi* natural forests and 15 *C. henryi* plantations of 3 cultivated varieties (TRY, TRC, and TRB) were analyzed using metagenomic sequences. An average of 10 G was yielded for each sample. After quality control and BLAST, 86.35, 0.80, and 0.50% of the sequence reads corresponded to the domains Bacteria, Fungi, and Archaea on the basis of taxonomic assignment of the microbial cell fraction (Appendix A). As can be seen from Figure 1, the Ace and Chao1 richness indices and the Shannon and Simpson diversity indexes were highest in TTX, and the order of species richness and diversity was TTX > TRC > TRB > TRY. The microbial diversity and richness of natural forest were significantly higher than that of plantation forest.

However, there are differences among the three cultivated varieties of plantation. We observed that the microbial diversity of TRC was significantly higher than other varieties, indicating that different cultivated varieties can affect the diversity of soil microbial community. Principal coordinate analysis (PCoA) is used to distinguish the community structure of soil microorganisms in plantation and natural forest. The PC1 and PC2 of PCoA explained 55.70 and 13.73% of overall variability of microbial community composition, respectively. The group of natural and the plantation forest were significantly distinguished. Two cultivated varieties (TRY and TRB) were relatively similar while significantly distinguished from another one (TRC) (Figure 2).

### 3.2. Specific Differences in Composition of Microbiome at Different Forest Types and Cultivated Varieties

Metagenome sequences showed that an average of 181 was annotated at the level of phyla. On different forest types, the dominant phyla of natural forest were *Proteobacteria* (29.6%), *Acidobacteria* (23.9%), *Actinobacteria* (23.2%), *Planctomycetes* (1.7%), and *Verrucomicrobia* (1.2%), and the dominant phyla of plantation were *Acidobacteria* (35.9%), *Proteobacteria* (20.7%), *Actinobacteria* (15.0%), *Verrucomicrobia* (3.0%), and *Chloroflexi* (2.9%) (Figure 3a). *Proteobacteria*, *Acidobacteria*, and *Actinobacteria* were the three dominant phyla which account for more than 72.9% in the *C. henryi* rhizosphere of natural and plantation forest (Figure 3a). In order to clarify the species composition differences between different forest types and cultivated varieties, the phyla with an average relative abundance greater than 0.01% in five duplicate samples were selected to compare the differences. We observed that *Proteobacteria* and *Actinobacteria* were significantly enriched in the natural forest and their average relative abundance was 8.9 and 8.2% higher than that in the plantation forest, respectively. However, *Acidobacteria* were massive and significantly enriched in the plantation and its relative abundance was 12.0% higher than that in the natural forest (Figure 3b). In addition, *Planctomycetes* was significantly enriched in the natural forest, while another nine phyla (mainly *Chloroflexi*, and *Verrucomicrobia*) were significantly enriched in the plantation. Although the number of enrichments in the natural forest was less than that in plantation, the abundance of phyla in plantation was mostly low, except for *Acidobacteria* (Figure 3b). On the different cultivated varieties, *Actinobacteria* and *Verrucomicrobia* were enriched in TRC and TRY, respectively, while *Thaumarchaeota* and *Gemmatimonadetes* show enrichment in TRB (Figure 3c). 

An average of 3183 was annotated at the level of genera and the dominant genus of the rhizosphere of *C. henryi* included *Bradyrhizobium*, *Streptomyces*, *Candidatus_Sulfopaludibacter*, and *Ktedonobacter* which were 3.9, 1.5, 0.8, and 0.8%. (Appendix A). The dominant genera of natural forest were *Bradyrhizobium* (6.7%), *Mycobacterium* (1.9%), *Streptomyces* (1.8%), and *Actinomadura* (1.0%). The dominant genera of plantation were *Bradyrhizobium* (3.0%), *Streptomyces* (1.5%), *Ktedonobacter* (1.0%), and *Candidatus_Sulfopaludibacter* (0.9%) (Appendix A). We observed that *Bradyrhizobium*, *Mycobacterium*, *Rhodoplanes*, and *Roseiarcus* were significantly enriched in the natural forest. Plantation mainly enriched *Ktedonobacter* and *Actinospica* (Appendix A). On different cultivated varieties, 43 genera were enriched at TRC, mainly including *Streptomyces*, *Actinospica,* and *Actinomadura*, while no genus was found to be enriched at TRY and TRB (Appendix A).

### 3.3. Relationship between Microbial Communities and Soil Physicochemical Properties

The physical and chemical properties of the tested soils were different (Table 1). Briefly, the soil pH, soil moisture, OM, TK, TC, and TN contents of natural forest were higher than those of artificial forest. There was no significant difference in soil physical and chemical properties among three cultivated varieties of plantation.

The heat map revealed that distinct soil nutrient has different effects on microorganism. The pH value and the TN content were positively correlated with *Actinobacteria* and *Proteobacteria* and negatively correlated with *Acidobacteria*, *Cyanobacteria,* and *Verrucomicrobia*. The OM content was positively correlated with *Basidiomycota* and negatively correlated with *Chloroflexi*, *Cyanobacteria,* and *Verrucomicrobia*. The SM content was generally negatively correlated with *Verrucomicrobia*. The TP content was positively correlated with *Actinobacteria* and negatively correlated with *Acidobacteria*. The TK content was generally negatively correlated with *Cyanobacteria* and *Verrucomicrobia*. The TC content was positively correlated with *Proteobacteria* and negatively correlated with *Cyanobacteria* and *Verrucomicrobia*. In sum, the bacteria enriched in natural forest were mostly positively correlated with soil physical and chemical properties, while the bacteria enriched in plantation forest were negatively correlated (Figure 4).

### 3.4. Specific Differences in Function of Microbiome at Different Forest Types and Different Cultivated Varieties

Functional annotations were obtained by blasting against the KEGG Orthology (KO) database (https://www.kegg.jp/kegg/ko.html, accessed on 24 April 2021). 8214, 7239, 5986, and 6119 KOs were annotated at TTX, TRC, TRB, and TRY, respectively. A total of 9003 KOs has mainly been involved in 4 KEGG level 1 pathways, 22 KEGG level 2 pathways, and 174 level 3 pathways (Figure 5a and Appendix A).

PCoA revealed that the KOs of the rhizosphere had significant differences between the natural forests and plantation (Appendix A). At the KEGG level 2 pathways, more than half were observed with significant enrichment in natural forests than plantation (Kruskal–Wallis rank-sum test, *p* < 0.05) (Appendix A). At the KEGG level 3 pathways, we observed that 83 level 3 pathways were significantly enriched in natural forest, while only 3 were enriched in the plantation (Wilcoxon rank-sum test, *p* < 0.05) (Figure 5b). The enriched pathway in the natural forests mainly corresponds to 4 pathways, which were metabolic pathways (15.2%), biosynthesis of secondary metabolites (7.0%), microbial metabolism in diverse environments (6.3%), and biosynthesis of antibiotics (5.4%). The enriched pathway in the plantation were RNA polymerase (0.4%), other glycan degradation (1.0%), and carbon fixation pathways in prokaryotes (0.5%) (Figure 6a). On the different cultivated varieties, 17 pathways were enriched in TRC and mainly included Quorum sensing (1.5%), ABC transporters (1.4%), Phenylalanine metabolism (0.4%), and degradation of aromatic compounds (0.3%). The pathway of Chloroalkane and chloroalkene degradation is enriched in TRB, which accounts for 0.35%. However, no pathway is enriched in TRY (Kruskal–Wallis rank-sum test, *p* < 0.05, Figure 6b).

### 3.5. Molecular Ecological Network Structure of Soil Microbial Community

There were 818 nodes and 2165 links in the rhizosphere soil network structure of the natural forest, with an average degree of 5.29, an average path distance of 9.97, and an average clustering coefficient of 0.42. The network of TRY, TRB, and TRC consisted of 817, 815, and 832 nodes and 1967, 1748, and 2021 links, with an average degree of 4.81, 4.29, and 4.85, an average path distance of 10.30, 14.02, and 13.07, and an average clustering coefficient of 0.44, 0.36, and 0.43, respectively (Appendix A). The number of negative correlation links in the natural forests was 593, accounting for 27% of the corresponding total links, while the negative correlation links in the plantation (TRY, TRB, and TRC) were 835, 622, and 686, accounting for 43, 36, and 34% of the corresponding total links, respectively (Appendix A). These results showed that the network structure in the natural forests was more complex and microorganisms in the network were mainly synergistic, while the network structure in the plantation was simpler and microorganisms in the network show strong competitiveness.

Then, we observed that *Proteobacteria, Actinobacteria,* and *Basidiomycota,* accounting for 75% of total nodes, were the dominant nodes in the natural forests and plantation network (Figure 7a).

However, *Acidobacteria,* with the average relative abundance of 23–43%, only accounts for less than 0.3% of the total nodes in the network. Venn’s analysis of nodes in the natural forests and plantation networks shows that up to 92% of nodes (817 out of 887) were shared, and only 1 and 70 nodes were unique in natural forests and plantation, respectively (Appendix A). Meanwhile, in three cultivated varieties, 80% of nodes (709 out of 887) were shared and only few nodes were unique in TRY, TRB, and TRC, respectively (Appendix A). According to the definition of abundance, the network of natural forest was constituted by 67.48% rare genera, 26.77% moderate genera, and only 5.75% abundant genera. The network of three cultivated varieties was constituted by more than 67% rare genera and 21% moderate genera, and only less than 6% abundant genera (Appendix A). Then, the top 10 nodes as keystones taxa in each network were selected according to the value of degree (from high to low). We found that 27 out of 40 key nodes belonged to rare and moderate genera, while only 13 nodes belonged to abundant genera (Figure 7b). These results showed that the abundance of a species was not the best determinant of its contribution to the community, although species with low abundance still play an important role.

## 4. Discussion

The long-term continuous cropping of plantation leads to the decline of forest productivity [32] and soil degradation [54], and the loss of soil microbial diversity is considered as the main threat to the balance of ecosystems [55]. Different forest planting methods can affect soil microbial communities and the change of microbial communities may affect the function of soil ecosystem [3,15]. In this research, we compared the differences of microbial diversity among rhizosphere soil of *C. henryi* forest cultivated in natural and plantation forests. Our findings clearly showed that the natural forests have higher diversity and richness at species composition and functional genes and the network relationship is more complex.

The soil organic matter of natural forest can be satisfied by the natural decomposition of abundant litter [56]. Plantations are often characterized by intensive single cultivation [32] and the return of litter is significantly lower than that of natural forest, whether at a young age or after maturity [56]. For instance, Yang et al. found that the content of soluble organic nitrogen in 0–10 cm soil decreases by 27.7% when the natural forest of *Castanopsis kawakamii* converts into three kinds of artificial forests [35]. Allen et al. found that the soil fertility, microbial biomass, and NH^4+^ conversion rate was gradually decreased when the forests were transformed into the two kinds of rubber plantations [57]. The main nutrient source of plantation soil is fertilization. Short-term and reasonable fertilization can effectively improve soil conditions, while long-term and unreasonable fertilization will imbalance soil nutrient elements and cause soil acidification [37,58,59]. According to statistics, in some major areas of China, over-application of chemical fertilizers has caused serious soil acidification and reduced soil pH value by 0.13 to 2.20 [59]. However, pH is one of the key factors that drives the microbial community construction and it can well predict the microbial diversity and richness, whether on a continental scale [60] or on different altitude gradients [61] and even on different plant compartments [62]. Meanwhile, due to the monoculture in *C. henryi* plantation and the frequent occurrence of serious diseases and pests, *C. henryi* plantation mainly depends on spraying pesticides for control. Low concentration and short-term application of chemical pesticides can well control diseases and pests, but the long-term excessive application will cause pesticide residues and environmental pollution [38,63]. This unreasonable operation caused soil hardening, nutrient imbalance, and soil erosion, thus reducing the diversity and abundance of microbial communities. Plant root exudates are also a key factor affecting the accumulation of soil microbial species [64]. Previous studies have shown that the plants with different genotypes can secrete different types and amounts of root exudates [65,66]. For instance, there are significant differences in specific compounds in the root exudates of the three soybean varieties, with the variety Nice-Mecha exuding about two times more organic acids than variety Bara, and the variety Svapa exuding more sugars than variety Bara [67]. Hence, we speculated that the differences among different cultivated varieties were very likely to be explained by the difference in root exudates.

Our study showed *Proteobacteria*, *Acidobacteria*, and *Actinobacteria* are the dominant flora in the rhizosphere soil of two *C. henryi* forest types, accounting for >72.9% of all the phyla. The abundance of *Proteobacteria* and *Actinobacteria* in rhizosphere soil of natural forest was higher than that of plantation, while the abundance of *Acidobacteria* in plantation was higher than that of natural forest. By analyzing the microbial communities of litter and soil of forests, Urbanova et al. found that *Proteobacteria* and *Actinobacteria* were major phyla, accounting for more than 50 and 13% both in litter and soil [26], suggesting that they are the main populations for the decomposition and transformation of forest lit-ter. *Actinobacteria* can decompose the refractory organic matter in the high carbon content soil [68], affecting the input of soil nutrients and the formation of soil aggregate structure. Meanwhile, *Actinobacteria* can participate in the decomposition of aromatic compounds and other complex compounds [69,70]. In addition, they are able to produce a wide spectrum of secondary metabolites, such as antibiotics and organic acid [71], to resist pathogenic microorganisms and harmful environmental microorganisms [71,72]. *Proteobacteria* is also an important flora for decomposing soil organic matter, fixing nitrogen and carbon, and dissolving phosphorus [26,73,74,75]. It has been reported that *Proteobacteria* can also degrade aromatic compounds to reduce the organic pollution [38]. Our research found that the carbon metabolism pathway, the biosynthesis of antibiotics pathway, and the biosynthesis of secondary metabolites pathway were enriched in the natural forest. Therefore, natural forests may increase the carbon metabolism pathways of microbial communities by enriching *Proteobacteria* and *Actinobacteria* to enhance the decomposition of understory litter and release nutrients to improve soil quality. At the same time, *Proteobacteria* and *Actinobacteria* also enhance the biosynthesis of secondary metabolites and antibiotics pathway to improve the ability of the host to resist the invasion of pathogens. *Acidobacteria* mostly exists in acidic environments, and the abundance decreased with pH value [76,77]. Meanwhile, it is reported that some *Acidobacteria* members can tolerate the environment of nutrient deficiency. Therefore, it can be used as an evaluation index of soil fertility [78]. Compared with the natural forest, the *C. henryi* plantation has lower pH and soil nutrients, as well as lower soil moisture (SM). Unreasonable fertilization and pesticide spraying in plantations leads to deterioration of soil quality, such as soil acidification and compaction and nutrient imbalance. Hence, the characteristic of a single planting structure and unreasonable management measures may cause the decline of plantation soil fertility. The degradation of soil may be the main reason for the accumulation of *Acidobacteria* in the plantation.

From our network analyses, we found that higher nodes, links, and average degree in the natural forests network indicate that the network of the natural forest is more complex than that in the plantation [79]. The interaction patterns in the natural forests network were identified as predominantly positive, meaning that microbial interactions tended to favor symbiosis rather than competition [80,81]. As indicated in previous studies, the positive correlations are dominant among rhizosphere bacteria which are usually rich in nutrients [17,79]. Li et al. found that the percentage of positive links of soil microbial communities in *Panicum virgatum* increased from 72.0 to 87.9% under three N fertilizer levels (0, 56, 196 kg ha^−1^) [17]. Hence, rich nutrients may alleviate the competition of microorganisms. However, we found more negative correlations existed in plantations. Studies on the social network structure of animals have shown that the availability of resources is one of the key drivers of network structures [82,83]. Similarly, finite nutrients can invoke a relentless war among diverse microorganisms [84]. We conjecture that poor plantation soil triggered microorganism competition. In our study, the top 10 nodes which have high connections (that is, degree) were selected as keystone taxa in each network and the removal of keystone taxa will lead to drastic changes in the composition and function of the microbiome [85,86]. We observed that the keystone taxa, whether in natural forest or plantation network, are mostly rare and moderate abundant taxa, consistent with Xiong et al. [52] and Xue et al. [87]. However, although the dominant taxa with high relative abundance are very important and can affect ecosystem functioning or a specific process, the taxa with low abundant taxa should not be neglected because they act as an important or potentially important role in maintaining microbial networks [88].

## 5. Conclusions

In summary, we found that the microbial diversity and richness of natural forest were significantly higher than that of plantation forest. *Acidobacteria*, *Proteobacteria*, and *Actinobacteria* are the three dominant phyla, and the abundance was affected by different forest types and cultivated varieties. *Proteobacteria* and *Actinobacteria* are enriched in natural forests, while *Acidobacteria* is massively enriched in the plantation. Meanwhile, we found that a more complex network structure and more positive interactions exist in natural forest than plantation. The keystone taxa are mostly rare and moderate abundant taxa. Our results provide a scientific basis for the sustainable management of the plantation. Furthermore, the follow-up work should further study the variation law and influencing factors of microbial community under seasonal and interannual differences for realizing more reasonable and effective agricultural management measures in the future.

## Figures and Tables

**Figure 1 microorganisms-10-00042-f001:**
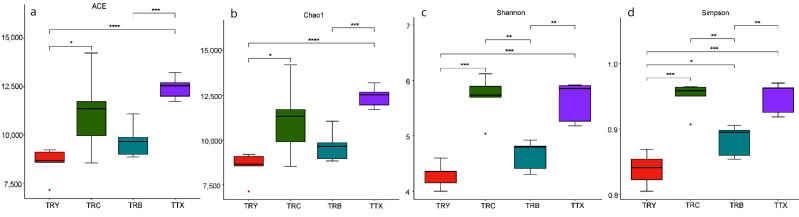
Alpha diversity of the rhizosphere microbiome of *C. henryi.* (**a**) ACE index, (**b**) Chao1 index, (**c**) Shannon index, (**d**) Simpson index. * Correlation is significant at *p* < 0.05; ** Correlation is significant at *p* < 0.01; *** Correlation is significant at *p* < 0.001. TRY, TRC, and TRB, three cultivated varieties of plantation; TTX, natural forest.

**Figure 2 microorganisms-10-00042-f002:**
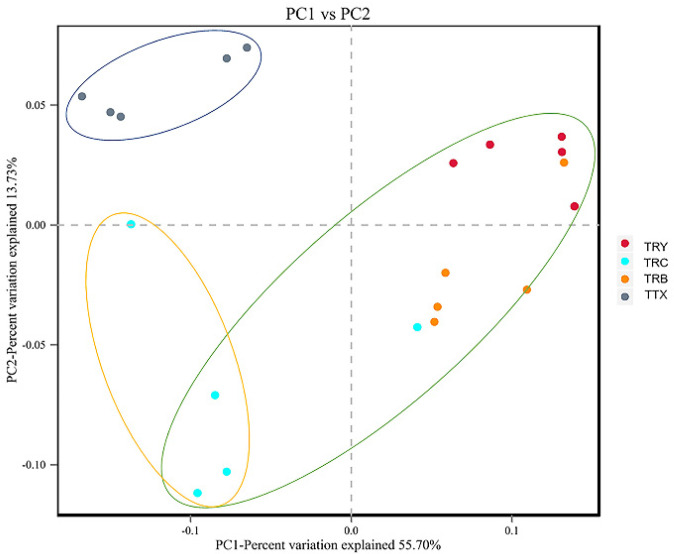
The principal coordinate analysis (PCoA) based on Bray-Curtis distance of rhizosphere microbiome in *C. henryi.*

**Figure 3 microorganisms-10-00042-f003:**
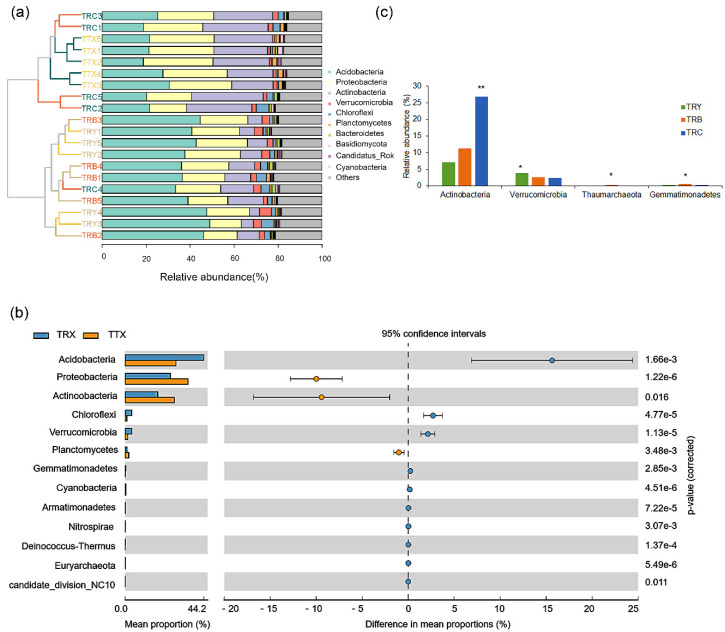
The relative abundance and differences of rhizosphere microbiome (at phylum) in *C. henryi*. (**a**) Relative abundance (%) of the major taxonomic distribution at Phylum. (**b**) The pairwise comparison difference at phyla were calculated with Welch’s *t*-test. (**c**) The multiple comparisons comparison difference at phyla were calculated with Kruskal–Wallis rank-sum test. * Correlation is significant at *p* < 0.05; ** Correlation is significant at *p* < 0.01; TRY, TRC, and TRB, three cultivated varieties of plantation; TTX, natural forest; TRX, plantation.

**Figure 4 microorganisms-10-00042-f004:**
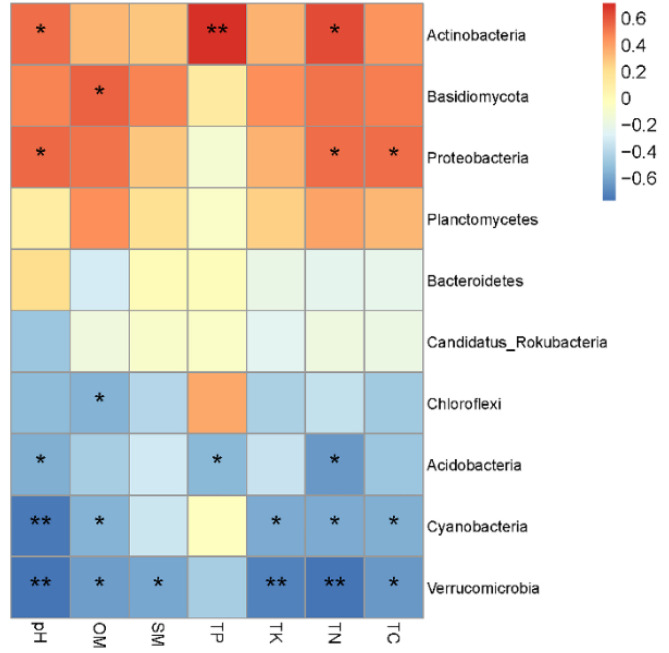
The Spearman correlation heatmap between top ten abundance microorganisms (phylum level) and soil physicochemical characteristics. The size of Spearman correlation coefficient is displayed by color gradient. * Correlation is significant at *p* < 0.05; ** Correlation is significant at *p* < 0.01; pH, soil pH; OM, organic matter; SM, soil moisture; TP, total phosphorus; TK, total potassium; TN, total nitrogen; TC, total carbon.

**Figure 5 microorganisms-10-00042-f005:**
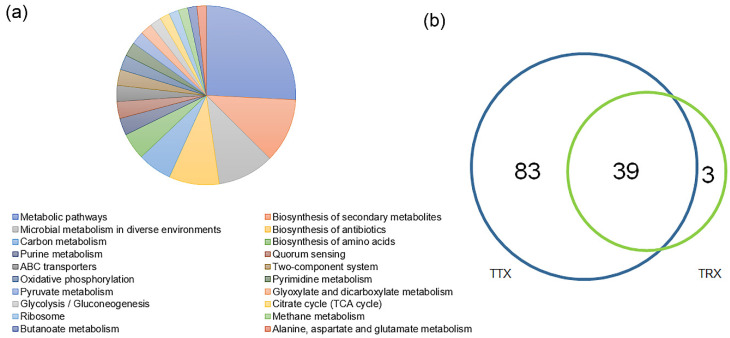
Characterization of the function of the *C. henryi* rhizosphere micro-biome. (**a**) Functional KEGG level 3 pathways of rhizosphere microbiomes of *C. henryi* and only the top 20 were shown. (**b**) Venn plot depicting the number of KEGG level 3 pathways based on the relative abundance (>0.01%) in all samples and the pathway enrichment between natural forests and plantation.

**Figure 6 microorganisms-10-00042-f006:**
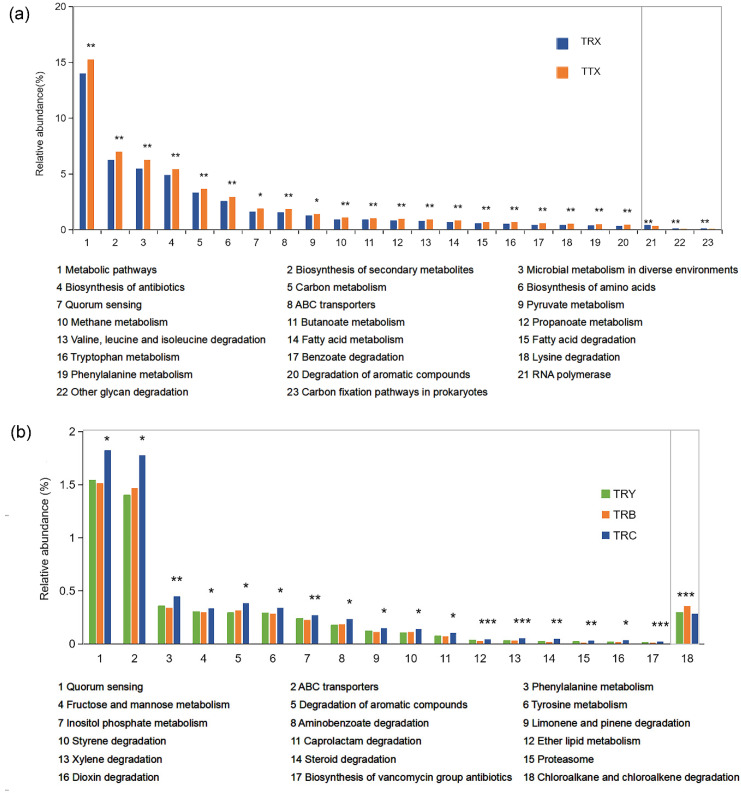
Differences functional composition of rhizosphere microbiome in *C. henryi*. (**a**) The pairwise comparison difference at KEGG level 3 pathways were calculated with Wilcoxon rank-sum test. TRX stands for plantation. (**b**) The multiple comparisons comparison difference at KEGG level 3 pathways were calculated with Kruskal–Wallis rank-sum test. * Correlation is significant at *p* < 0.05; ** Correlation is significant at *p* < 0.01; *** Correlation is significant at *p* < 0.001. TRY, TRC, and TRB, three cultivated varieties of plantation; TTX, natural forest; TRX, plantation.

**Figure 7 microorganisms-10-00042-f007:**
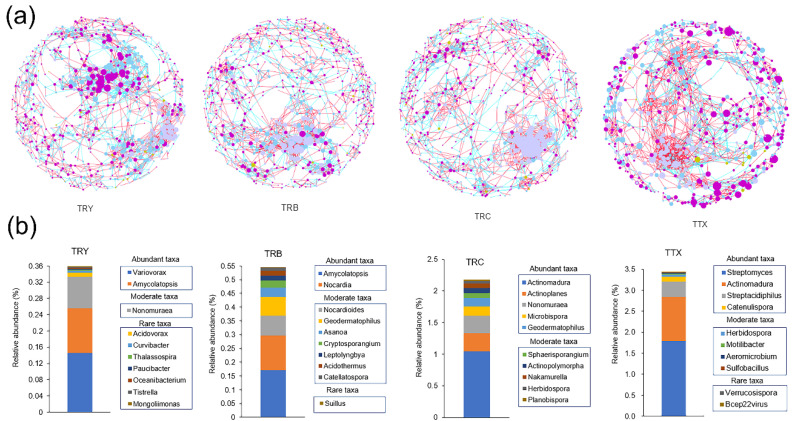
Taxonomic network inference of the rhizosphere microbiome. (**a**) The molecular ecological network was calculated by MENA and visualized by Cytoscape. Different nodes represent different genus. The nodes belonging to *Proteobacteria*, *Actinobacteria*, and *Acidobacteria* members are colored in dark purple, light purple, and yellow-green, respectively. Other nodes are uniformly colored blue. Blue edges represent negative correlation and red edges represent positive correlation. Size of each node is proportional to the number of connections (that is, degree). (**b**) The relative abundance of top 10 key nodes.

**Table 1 microorganisms-10-00042-t001:** Physical and chemical properties of the *C. henryi* rhizosphere soils.

Sample	TRY	TRB	TRC	TTX
pH	4.26 ± 0.40 b	4.55 ± 0.12 ab	4.41 ± 0.13 ab	4.82 ± 0.13 a
OM (g/kg)	29.99 ± 4.25 b	30.80 ± 4.43 b	32.75 ± 3.86 b	54.18 ± 7.25 a
TP (g/kg)	0.10 ± 0.03 c	0.24 ± 0.02 a	0.20 ± 0.04 ab	0.18 ± 0.03 b
TK (g/kg)	6.19 ± 2.44 b	8.37 ± 1.05 b	8.39 ± 1.73 b	22.94 ± 5.69 a
TC (g/kg)	15.73 ± 3.08 b	17.61 ± 2.08 b	16.76 ± 1.83 b	29.02 ± 2.85 a
TN (g/kg)	1.31 ± 0.13 b	1.60 ± 0.16 b	1.45 ± 0.13 b	2.03 ± 0.14 a
SM (g/kg)	202.66 ± 22.75 b	215.11 ± 32.36 b	259.59 ± 35.88 b	283.71 ± 36.15 a

Significant test at *p* < 0.05. pH, soil pH; OM, organic matter; TP, total phosphorus; TK, total potassium; TC, total carbon; TN, total nitrogen; SM, soil moisture. The same lowercase letters following the means indicate non-significant differences between corresponding treatments. TRY, TRC, and TRB, three cultivated varieties of plantation; TTX, natural forest.

## Data Availability

The raw sequencing reads were deposited in the NCBI Bioproject database under the accession number PRJNA779280.

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
