# Peer review of "Deciphering Rhizosphere Microbiome Assembly of Castanea henryi in Plantation and Natural Forest"

_microorganisms, 2021, doi:10.3390/microorganisms10010042_

Round 1

Reviewer 1 Report

The authors of the manuscript explore the changes of microbial communities between natural and plantation Castanea henryi forests using the metagenome sequencing. This manuscript contains a lot of important findings, and the analyses are appropriate. However, I have two major concerns as follows:

(1) Explanations on the sampling methods are insufficient. Although the authors described “Five standard sample plots are randomly set at each of the four sampling points.”, I don't know how big each plot and each sampling point are, or how far apart are between plots and between sampling points. More importantly, I am wondering whether this sampling method was actually pseudoreplication in ecological field experiments. In addition, I don't understand the definition of rhizosphere soil and how to collect it.

(2) There was little discussion on the effect of three different cultivated varieties on soil properties and microbial community structures. In addition, there was no information on management practices such as fertilization and pesticide spraying for plantations. It seems that fertilization caused lower pH in plantation than in natural forest. Therefore, I am not sure whether different soil properties and microbial communities would be caused by difference in root exudation among different varieties (L350-352) or different fertilization treatments.

Other comments:

L45, what is “AMF”?

L64, what is “andorganic”? Please double-check.

L83, what does “single structure and management of the plantation” mean? Please re-write the sentence.

L106-108, Please explain more details and cite the references, if necessary, how to measure TP and TK.

L205, is “at phyla” OK?

L276 and L299, I don't think the titles are suitable for the result section.

Table S1, how about show Table S1 not an appendix but as Table 1, since the data in Table S1 were essential for the study. In addition, the table should be self-explained and be able to stand alone, and thus, the description should be complete. So, please describe what are a, b, c, and TRY, TRB, OM and so on. Finally, were a, b, bc and c in TP correct?

Reviewer 2 Report

In my opinion, the article is interesting and can be accepted for publication after editing the manuscript. My comments are below.

Line 32-33 Omit the text „pathogen control“  This makes no sense in the sentence.

Line 39-40 Omit this sentence.

Line 317-325 Any citation is missing. Please add. It is also worth thinking about this part of the text, because the statements in it do not always apply. My personal experience.

Line 326-329 Any citation is missing. Please add.

Line 357 What studies? Please add examples of studies.

Line 390 What studies? Please add examples of studies.

Line 393 What studies? Please add examples of studies.

Line 397 What studies? Please add examples of studies.

Line 400 What studies? Please add examples of studies.

Figure S2  The chart is probably not complete. What is 100% missing?

Round 2

Reviewer 1 Report

My concerns have been clarified. The revised manuscript can be accepted.